# Endocannabinoid Anandamide Attenuates Acute Respiratory Distress Syndrome through Modulation of Microbiome in the Gut-Lung Axis

**DOI:** 10.3390/cells10123305

**Published:** 2021-11-25

**Authors:** Muthanna Sultan, Kiesha Wilson, Osama A. Abdulla, Philip Brandon Busbee, Alina Hall, Taylor Carter, Narendra Singh, Saurabh Chatterjee, Prakash Nagarkatti, Mitzi Nagarkatti

**Affiliations:** 1Department of Pathology, Microbiology and Immunology, School of Medicine, University of South Carolina School of Medicine, Columbia, SC 29209, USA; muthanna.sultan@uscmed.sc.edu (M.S.); kiesha.wilson@uscmed.sc.edu (K.W.); osama.abdulla@uscmed.sc.edu (O.A.A.); Brandon.Busbee@uscmed.sc.edu (P.B.B.); Alina.hall@uscmed.sc.edu (A.H.); taylor.carter@uscmed.sc.edu (T.C.); narendra.singh@uscmed.sc.edu (N.S.); PRAKASH@mailbox.sc.edu (P.N.); 2Environmental Health and Disease Laboratory, Arnold School of Public Health, University of South Carolina, Columbia, SC 29208, USA; schatt@mailbox.sc.edu

**Keywords:** acute respiratory distress syndrome (ARDS), Staphylococcus enterotoxin B (SEB), anandamide (AEA), COVID-19, microbiome, MiSeq sequencing, gut-lung axis, antimicrobial peptides (AMPs), single-cell RNA (Sc-RNA), short-chain fatty acids (SCFAs)

## Abstract

Acute respiratory distress syndrome (ARDS) is a serious lung condition characterized by severe hypoxemia leading to limitations of oxygen needed for lung function. In this study, we investigated the effect of anandamide (AEA), an endogenous cannabinoid, on Staphylococcal enterotoxin B (SEB)-mediated ARDS in female mice. Single-cell RNA sequencing data showed that the lung epithelial cells from AEA-treated mice showed increased levels of antimicrobial peptides (AMPs) and tight junction proteins. MiSeq sequencing data on 16S RNA and LEfSe analysis demonstrated that SEB caused significant alterations in the microbiota, with increases in pathogenic bacteria in both the lungs and the gut, while treatment with AEA reversed this effect and induced beneficial bacteria. AEA treatment suppressed inflammation both in the lungs as well as gut-associated mesenteric lymph nodes (MLNs). AEA triggered several bacterial species that produced increased levels of short-chain fatty acids (SCFAs), including butyrate. Furthermore, administration of butyrate alone could attenuate SEB-mediated ARDS. Taken together, our data indicate that AEA treatment attenuates SEB-mediated ARDS by suppressing inflammation and preventing dysbiosis, both in the lungs and the gut, through the induction of AMPs, tight junction proteins, and SCFAs that stabilize the gut-lung microbial axis driving immune homeostasis.

## 1. Introduction

ARDS is defined as a form of respiratory failure that is caused by a variety of insults, including pneumonia, sepsis, trauma, and certain viral infections [1,2]. One of the common features of ARDS is the hyperactivation of the immune response, systemically and in the lungs, leading to the development of pulmonary edema, alveolar damage, and respiratory failure [3]. ARDS is considered to be a major worldwide health problem in the field of clinical respiratory medicine [4]. ARDS affects approximately 200,000 patients every year in the United States and causes over 75,000 deaths annually [5,6]. Thus, it is difficult to treat ARDS, and nearly 37% of ARDS patients die annually. Globally, 10% of intensive care unit admissions represent ARDS cases, which accounts for over 3 million patients annually [7].

Coronavirus disease 2019 (COVID-19) caused by SARS-CoV2 has triggered the global pandemic with more than 209 million infections and over 4 million people killed to date. People who contract the severe form of COVID-19 go on to develop ARDS and manifest a systemic hyperimmune response with a cytokine storm which is difficult to treat, leading to high rates of mortality. While the precise cause of hyperimmune response is unclear, there is evidence to support the possibility that hyperactivation of the immune response may result from alterations in microbiota and advancement of secondary infections. The prevalence of coinfection can account for up to 50% of patients who die from COVID-19 [8]. Coinfection can result from pathogens, such as *Streptococcus pneumoniae*, *Staphylococcus aureus*, and *Klebsiella pneumoniae* [8]. *Staphylococcus* produces Staphylococcus enterotoxin B (SEB), which acts as a superantigen, thereby activating a large proportion of T cells, causing cytokine storm, ARDS, multiorgan failure, and often death [6,9,10]. Staphylococcus enterotoxins produced by *Staphylococcus aureus* are also known to cause food poisoning and toxic shock syndrome [11]. Staphylococcus enterotoxin B (SEB) is classified as a biological threat agent due to its ability to cause fatal toxic effects [12]. Our laboratory has investigated multiple murine models of SEB-mediated ARDS and found that cannabinoids are highly effective in attenuating ARDS [9,13,14].

Cannabinoids are compounds that activate two types of G-protein-coupled receptors, CB1 and CB2, that are expressed primarily in the CNS and cells of the immune system, respectively [15,16]. They are found naturally produced in the body (endocannabinoids) or in the cannabis plant (phytocannabinoids). The cannabis plant has over 120 phytocannabinoids, of which Δ9-tetrahydrocannabinols (THC) is very well characterized for its ability to activate CB1 and thereby mediate psychoactive properties. THC can also activate the CB2 receptors expressed on immune cells and mediate anti-inflammatory effects [17,18]. The endocannabinoids, which are host-derived lipid hormones, are found in all tissues, organs, and bodily fluids, and control a wide range of physiological functions, including appetite, pain, sleep, mood, and memory [19,20]. The two most well-studied endocannabinoids are the arachidonic acid derivatives, *N*-arachidonoylethanolamine (AEA) and 2-arachidonoylglycerol (2-AG). We and others have shown that endocannabinoids can also act as potent anti-inflammatory agents [21,22,23,24,25]. Recently, we demonstrated that AEA can attenuate SEB-mediated ARDS in a mouse model by targeting miRNA which triggers anti-inflammatory pathways involving immunosuppressive cells, such as the myeloid-derived suppressor cells (MDSCs) and the regulatory T cells (Tregs) [9]. Recent studies have shown that the diversity of the gut microbiota may be regulated by endocannabinoids [19,26]. This raises an important question of whether endocannabinoids can alter the microbiota in the gut and the lungs during ARDS, thereby providing increased protection from pathogenic bacteria as well as suppressing hyperinflammation. In the current study, we tested this hypothesis and our results demonstrated that AEA-mediated attenuation of SEB-induced ARDS may result from causing an increased abundance of beneficial bacteria that produce SCFAs that are anti-inflammatory and by inducing AMPs and tight junction proteins that prevent the emergence of pathogenic bacteria. The current study also provides evidence on the crosstalk of the microbiota in the gut-lung axis during ARDS. 

## 2. Materials and Methods

### 2.1. Mice Housing and Grouping

Female C57BL/6 mice, 6–8 weeks old, were purchased from Jackson laboratories. We performed our studies on females because it has been shown that females are more likely to develop acute respiratory distress syndrome [27]. Mice were housed at a density of no more than five animals per cage. Mice were kept under pathogen-free conditions in the Animal Resource Facility (ARF) at the University of South Carolina School of Medicine. These studies were approved by the Institutional Animal Care and Use Committee (IACUC). Mice were housed under a 12-h light/dark cycle at 18–23 °C and 40–60% humidity. The mice were randomized and then assigned for specific treatments.

### 2.2. Induction of SEB-Mediated ARDS in Mice and Treatment with AEA

ARDS was induced in mice as described previously [9,28]. Mice were given a single dose of SEB (Toxin Technology, Sarasota, FL, USA) intranasally at a concentration of 50 µg/mouse in 25 µL of phosphate buffer aaline (PBS) on day 0. On day-1, AEA (40 mg/kg body weight) or a vehicle (VEH) was given into these mice by the intraperitoneal (I.P.) route. AEA was dissolved in ethanol (50 mg/mL) and diluted further in PBS. Each mouse received 0.1 mL consisting of 84 μL of PBS and 16 μL of ethanol containing AEA. The vehicle controls (VEH) received 0.1 mL consisting of 84 μL of PBS and 16 μL of ethanol without AEA. The dose of AEA was based on our previous studies demonstrating that 40 mg/kg body weight of AEA attenuated T cell-mediated delayed-type hypersensitivity response and SEB-mediated ARDS [22]. The treatment with AEA was repeated on day 0 (SEB exposure day) and on day 1. Mice were euthanized on day 2 (48 h after SEB exposure) for various studies. Anandamide was purchased from Cayman chemicals (Ann Arbor, MI, USA).

### 2.3. Evaluation of Lung Function in Mice

To evaluate the effect of AEA on the functions of the lung, the whole-body plethysmography (Buxco, Troy, NY, USA) was used as described previously [29]. In brief, mice from each group were first restrained, in a plethysmography tube and were allowed to acclimatize. The clinical parameters of the lung, including peak inspiratory flow (PIF), peak expiratory flow (PEF), and tidal volume (TV) were evaluated. 

### 2.4. Collection of Serum and Broncho Alveolar Lavage Fluid (BALF)

To collect blood, mice were kept under anesthesia and then were bled through the retroorbital orifice. Bronchoalveolar lavage fluid (BALF) was collected as previously described [30], Briefly, bronchoalveolar lavage fluid was collected 48 h post-SEB exposure, after the mice were euthanized. The lung was excised as an intact unit along with the trachea. To collect BALF, sterile ice-cold PBS was injected through the trachea. 

### 2.5. Lung and Colon Histopathology

The lung and colon were excised after perfusion and the tissues were kept directly in 10% formalin followed by embedding in paraffin. The sections of lung and colon were cut in the core facility and then processed for hematoxylin and eosin (H&E) staining, as described previously [9]. Briefly, the lung and colon tissues were first mounted on glass slides and then the slides were transferred into xylene to deparaffinize the tissue sections. The tissue sections were then rehydrated in alcohol (100%, 95%, and 90%). The sections were finally stained with H&E, followed by dehydration. H&E-stained sections were analyzed using a KEYENCE (IL-US) digital microscope VHX-7000 (Itaska, IL, USA). Histopathological scoring parameters for lung and colon were evaluated as previously described [31,32].

### 2.6. Capillary Leak Measurement 

Evans blue assay was used to measure the capillary leak, as described previously [33]. Mice were injected intravenously with 1% Evans blue dye in PBS. The mice were sacrificed under general anesthesia after 2 h. The lungs were removed after perfusion with heparinized PBS and transferred to formamide and kept at 37 °C for 24 h. The calculation of leaked dye was measured by determining the absorbance of supernatant at 620 nm and then the following equation was applied to calculate the percentage increase in the capillary leak: (OD sample − OD control)/OD control) × 10.

### 2.7. Gut Leakage Measurement

Gut permeability in vivo was measured as described previously [34]. Briefly, C57BL/6 mice were administered FITC dextran (Sigma-Aldrich St Louis, Missouri-USA) dissolved in 100 μL of PBS by oral gavage, then, four hours later, blood was collected from the mice and the concentration of FITC-dextran was determined using a spectrophotometer (PerkinElmer Life Science, Akron, OH, USA) with an excitation wavelength of 480 nm.

### 2.8. Isolation of Mesenteric Lymph Nodes (MLNs) and Flow Cytometric Analysis

Mesenteric lymph nodes (MLNs) were isolated from euthanized mice and single cells were prepared in complete medium following squeezing of the tissue through a Stomacher 80 Biomaster blender (Steward, FL, USA). The cells were filtered using 70 μm filter and centrifuged for 10 min, at 1300 rpm. The pellet was resuspended in complete medium, and the cells were counted using a Bio-Rad TC 20 automated cell counter (Herculus, CA, USA).

The MLN cells were stained using fluorescent conjugated antibodies (anti-CD3 Brilliant Violet 785, anti-CD4 Phycoerythrin (PE), anti-CD8 Alexa Fluor 700, anti-Vβ8 FITC (Fluorescein isothiocyanate), and anti-NK1.1-PE/Dazzle. The staining of cells for dual markers was performed as described previously [35]. In brief, Fc receptor block was added to the MLN cells which were incubated for 10 min at room temperature, followed by staining of the cells using appropriate antibodies then incubation at 4 °C for 20–30 min. Stained cells were washed twice with cold PBS containing 2% fetal bovine serum (FBS) staining buffer.

### 2.9. Single-Cell RNA Sequencing (scRNA-seq) and Analysis 

scRNA-seq was performed as describe previously [36]. The whole lung tissue from mice was smashed using a Stomacher 80 Biomaster blender (Steward, FL, USA). The tissues were then filtered with a 70 µm filter twice and centrifuged at 1300 RPM for 7 min. Cell pellets were treated with RBC lysis on ice for five minutes, then washed twice in 10× buffer. Dead cells were removed using a Stem Cell Technologies dead cell removal kit (STEM CELL, Sunrise, FL, USA) and cell viability was measured using a TC 20 automated cell counter. The cell viability was 92–95%. A target number of 3000 cells from each group was then loaded into the Chromium Controller (10× Genomics) and 10× Genomics was used to process the cells. Following the manufacturer’s protocol, the chromium single cell 5′ reagent kits (10× Genomics) were used to process samples into single-cell RNA-seq (scRNA-seq) libraries. The sequencing of these libraries was performed using the NextSeq 550 instrument (Illumina) with a depth of 20–50 k reads per cell. The base call files generated from sequencing the libraries were then processed in the 10× Genomics Cell Ranger pipeline (version 2.0, San Francesco, CA, USA) to create FASTQ files. The FASTQ files were then aligned to the mouse genome, and the read count for each gene in each cell was generated. Each group was aggregated using the aggr pipeline in Cell Ranger, and the output was browsed in Loupe Browser 4.2.0. This allowed us to identify clusters and illustrated the differentially expressed genes between each cluster and groups.

### 2.10. MiSeq Sequencing of the Gut–Lung Axis 

First, microbial DNA from the cecal flush was isolated as described previously [37,38]. In brief, genomic DNA from the colon samples were isolated using a QIAamp DNA Stool Mini Kit, following the manufacturer’s protocol (Qiagen, Germantown, MD, USA). DNA from lung tissue was isolated using ZR Bacterial DNA MidiPrep (Zymo research, Irvine, CA, USA) and following the manufacturer’s instructions. The concentration of DNA was determined using a NanoDrop instrument (Thermo Fisher Scientific, Waltham, MA, USA). The isolated DNA was either used immediately or stored at −20 °C for future use. The 16S rRNA sequencing was performed on V3–V4 hypervariable regions of bacterial DNA of both lungs and gut. Amplification of DNA was performed, and Illumina adapter overhang nucleotide sequences (San Diego, CA, USA) were added. The sequencing was then performed on the Illumina MiSeq Platform as described previously [38]. For microbiome studies, operational taxonomic units (OTUs), alpha diversity (e.g., PD whole tree), and beta diversity (principal component analysis, or PCA) data were generated using the online National Institutes of Health (NIH)-based Nephele system (https://nephele.niaid.nih.gov/, accessed on 15 May 2021), using standard default settings for the 16S metagenomics pipeline. Linear discrimination analysis effect size (LEfSe) on Nephele-generated OTUs files was performed using the Huttenhower Galaxy web application. LEfSe data settings were performed using default settings as follows: the alpha factorial Kruskal–Wallis test among classes was set to 0.05, and the threshold on the logarithmic LDA score for discriminative features was set at 3. 

### 2.11. Quantitative Real-Time Polymerase Chain Reaction (qRT-PCR)

RT-qPCR was performed as described previously [9,39]. Briefly, cDNA was synthesized from total RNAs using the miScript II RT kit from Qiagen and following the protocol of the Qiagen company (Germantown, MD, USA). The details of primers used are presented in Table 1. 

### 2.12. Identification and Quantification of Short Chain Fatty Acids (SCFA)

Quantification of SCFAs was performed as described previously [40]. Briefly, 100 mg of cecal content was collected and then acidified with metaphosphoric acid and kept on ice for 30 min. Centrifugation at 12,000× *g* for 15 min at 4 °C was performed and then supernatant was collected and filtered using MC filters at 12,000× *g* for 4 min at 4 °C. The filtered samples were then transferred into glass vials and methyl tert butyl ether (MTBE) purchased from Sigma-Aldrich (St. Louis, MO, United States) was added. The samples were then centrifuged at 1300 rpm for 5 min at RT. The top organic layer was transferred to a new vial. The standard mixtures and internal standards were used to detect the response factors and linearity for each SCFA standard acid. To perform the analysis of SCFAs, HP 5890 gas chromatography configured with a flame-ionization detector (GC-FID) was used and 0.1 mM 2-ethyl butyric acid was used as an internal standard (IS) for all samples and standards. To quantify the concentration of SCFA, the Varian MS workstation (version 6.9.2) was used. 

### 2.13. Pretreatment with Butyrate and Treatment with Staphylococcus Enterotoxin B (SEB)

Mice were treated with butyrate or a vehicle (PBS) through oral gavage on day-1 at a dose of 200 mg/kg. On day 0, the mice were given another dose of butyrate with the same dose and 3 h later the mice received 50 μg SEB in 25 μL of PBS intranasally. On day 1, these mice received another dose of butyrate or the vehicle according to their respective groups. The mice were then euthanized on day 2 (48 h after SEB exposure) for further analysis of ARDS. The dose was based on our previous studies [35].

### 2.14. Statistical Analysis

Groups of three to seven mice were used in each experiment to study various aspects of ARDS, based on power analysis. The number of mice used in each experiment has been identified in the figure legends. Because of the smaller sample size in some experiments, we repeated the experiments three times for consistency, as detailed in the figure legends. For microbiome and SCFA screening studies, we used three mice per group based on our current studies showing distinct clusterings of mice within a group, as also reported previously by us [34,41]. We used GraphPad Prism Software (San Diego, CA, USA) for the statistical analyses. A Student’s *t*-test was applied to compare two groups, while one-way ANOVA with a post hoc Tukey’s test was used to compare three or more groups for most studies. For LEfSe data, the alpha factorial Kruskal–Wallis test among classes was set to 0.05, and the threshold on the logarithmic LDA score for discriminative features was set at 3. Data were expressed as mean ± SEM and statistically significant differences are presented in the figures as * *p* < 0.05, ** *p* < 0.01, *** *p* < 0.001, **** *p* < 0.0001.

## 3. Results

### 3.1. Anandamide Attenuates SEB-Mediated Inflammation in the Lungs 

Recently, we reported that AEA attenuates SEB-mediated acute lung injury [9]. The goal of this study was to explore the role of microbiota in this model, and to that end, we first performed experiments that corroborated our previous findings that AEA can suppress inflammation in the lungs and improve lung functions following exposure to SEB. Mice were exposed to SEB + Veh or SEB + AEA, as described in the Methods section, and 48 h later the lung functions were evaluated using the Buxco Instrument System. Peak inspiratory flow, peak expiratory flow, and tidal volume were measured. All parameters significantly improved in SEB + AEA, when compared to SEB + VEH (Figure 1A–C). These data demonstrated the ability of Anandamide to improve lung functions affected by SEB exposure. Next, we analyzed for the expression of IL-6, a key cytokine induced during ARDS by SEB [36], and found that it was highly induced following SEB treatment and was significantly attenuated following treatment with AEA both in the serum (Figure 1D) and BALF (Figure 1E).

### 3.2. Effect of Anandamide on the Lung and Colon in SEB-Exposed Mice 

Because SEB causes systemic cytokine storm and pathogenesis, we tested the effect of anandamide (AEA) on the lung and colon in SEB-exposed mice. Histopathological analysis and scores of the lungs demonstrated that AEA decreased the infiltration of inflammatory cells in the SEB + AEA group, when compared to the SEB + VEH group (Figure 2A). Interestingly, data obtained from the colon demonstrated that SEB exposure caused significant infiltration of inflammatory cells in the colon of the SEB + VEH group when compared to the Naïve group. AEA treatment decreased the SEB-mediated infiltration of the inflammatory cells in the colon (Figure 2B). 

Upon analysis of vascular and gut leakage, we noted that anandamide significantly decreased the leakage in the lungs (Figure 2D) and the gut (Figure 2E) in the SEB + AEA group, when compared to the SEB + VEH group. 

### 3.3. Anandamide Attenuates SEB-Activated T Cells in the Mesenteric Lymph Nodes (MLNs)

To further investigate the effect of AEA on gut-associated immune responses, we analyzed the effect of AEA on the mesenteric lymph nodes (MLNs). The data showed that AEA significantly decreased the percentage and absolute numbers of CD4 + T cells, CD8 + T cells, Vβ8 + T cells, and NKT cells in the SEB + AEA group when compared to the SEB + VEH group of mice (Figure 3A–D).

### 3.4. scRNA-seq of Cells Isolated from the Lungs 

Using scRNA-seq, we first looked at the different types of cell clusters in the lungs of SEB + VEH and SEB + AEA mice. As shown in Figure 4A, we detected 12 different types of cells isolated from the lungs and the criteria used is detailed in the figure’s legend. The data showed that some of the cell clusters were different between the SEB + VEH vs. the SEB + AEA group. Of significant note, there was an increase in the cluster numbered 6 (regulatory cell 2) in SEB + AEA when compared to SEB + VEH, consistent with our previous studies that AEA induces Tregs [9]. In addition, there was a significant decrease in the clusters numbered 8 and 9 (macrophages 2 and 3, respectively) in the SEB + AEA group when compared to SEB + VEH. Lung epithelial cells play an important role, such as barrier protection, fluid balance, clearance of particulate matter, and protection against infection [42]. Furthermore, they produce antimicrobial peptides (AMPs), which act as antibiotics to kill pathogenic bacteria [43]. We therefore performed scRNA-seq of epithelial cells (cluster 10) from the lungs to study the expression of AMPs. The data demonstrated that AEA significantly upregulated the expression of AMP-related genes in the lung epithelial cells (Figure 4B). The AMPs-related genes, including tracheal antimicrobial peptide (TAP1) (Figure 4B), tracheal antimicrobial peptide 2 (TAP2) (Figure 4C), lysozyme 2 (LYZ 2) (Figure 4D), and murine beta defensin 2 (MBD2) (Figure 4H). We also noted increased expression of tight junction proteins such as claudin 1 (CLDN1) (Figure 4F) and cadherin 1 (CDH1) (Figure 4G), as well as secretory leukocyte peptidase inhibitor (SLPI) (Figure 4E) in the SEB + AEA group, in contrast to the SEB + VEH group. Moreover, these data were confirmed by performing RT-qPCR (Figure 4B–H). 

### 3.5. The Effect of AEA on Microbial Profiles in the Lungs and the Gut

Because we noted significant induction of AMPs following AEA treatment, we next investigated the microbial profile in the lungs and the gut following SEB treatment. SEB treatment increased the abundance of microbiota in the lungs, and AEA significantly reversed this, as shown using Chao1 rare fraction measure (Figure 5A). Beta diversity analysis, which measured the similarity or dissimilarity between various groups, showed that all three groups were well separated, with the SEB + AEA group clustered away from the SEB + VEH group (Figure 5B). Additionally, representatives of the orders of Caulobacterales and Pseudomonodales were significantly decreased in the SEB + AEA group compared to the SEB + VEH (Figure 5C,D). To distinguish significantly altered bacteria among the three groups, linear discriminant analysis effect size (LEfSe) analysis (Figure 6A) was performed and the corresponding cladogram (Figure 6B) was generated. Data showed that there were several bacteria found to be distinctly expressed in each of the three groups. Our LEfSe analysis of the lungs indicated that beneficial bacteria, such as Muribaculaceae (s24-7), was indicated in the SEB + AEA group only. Additionally, some beneficial bacteria, such as Lachnospiraceae and Clostridia which produce butyrate, were indicated in the SEB + AEA group but not in the Naïve or the SEB + VEH group. Interestingly, some pathogenic bacteria such as Pseudomonas and Enterobacteriaceae were indicated in the SEB + VEH group only but not in the Naïve or SEB + AEA. 

### 3.6. Role of Anandamide in the Regulation of Microbial Dysbiosis in the Gut

Previously we have shown that there is crosstalk between the gut and lung microbiota which regulates ARDS [37]. To that end, we looked to see whether SEB caused similar alterations in the gut and if AEA would reverse these changes. SEB treatment decreased the abundance of microbiota in the gut and AEA failed to reverse this, as shown using Chao1 rare fraction measure (Figure 7A). Beta diversity analysis showed that all three groups were well separated, with the SEB + AEA group clustered away from the SEB + VEH group (Figure 7B). The analysis of microbiota using MiSeq of 16S RNA of microbiota of the gut indicated that Anaeroplasmatacae at the family level and Tenericutes at the phylum level were significantly increased in the SEB + VEH group compared to the naïve mice, and treatment with AEA reversed this change (Figure 7C,D). To distinguish significantly altered bacteria among the three groups, LEfSe analysis (Figure 8A) was performed and the corresponding cladogram (Figure 8B) was generated. Our LEfSe analysis of the gut indicated that Muribaculaceae (S24-7) was indicated in the gut of the SEB + AEA group. Anaeroplasma and Tenericutes, which are pathogenic bacteria, were seen in the gut of the SEB + VEH group only but not in the Naïve or SEB + AEA group. These data showed that there were distinct bacteria found to be uniquely expressed in each of the three groups. 

### 3.7. Effect of Anandamide on Short Chain Fatty Acids (SCFAs) in the Gut

Gut microbiota are known to produce SCFAs that suppress inflammation [44,45]. To that end, we measured the levels of SCFAs in Naïve, SEB + VEH and SEB + AEA and we found that butyric acid, valeric acid and isovaleric acid levels were significantly decreased in the SEB + VEH group when compared to the naïve mice, while in the SEB + AEA group, the levels of these SCFAs increased significantly compared to the SEB + VEH group (Figure 9).

### 3.8. The Role of Butyrate in the Amelioration of SEB-Mediated ARDS

We next tested the role of butyrate in attenuating ARDS by directly administering it into SEB-injected mice. The data showed that administration of butyrate significantly improved the SEB-mediated ARDS, as seen from the decreased infiltration of inflammatory cells and histological scores in the lungs (Figure 10A). The butyrate also improved the clinical lung parameters, such as specific airways resistance (Figure 10B) and specific airways conductance (Figure 10C).

## 4. Discussion

ARDS resulting from a cytokine storm is difficult to treat and, thus, there is dire need to develop new treatment modalities. In this study, we investigated the role of endocannabinoid, AEA, in the attenuation of SEB-mediated ARDS, and found that AEA was effective in ameliorating ARDS. For the for the first time, we identified unique pathways through which AEA may mediate this effect. Primarily, AEA was found to modulate the microbiome profile in the gut-lung axis and suppress inflammation through production of AMPs and tight junction proteins expressed in lung epithelial cells. The expression of these molecules was detected through Sc-RNA sequencing and revealed that AEA may promote beneficial bacteria that produce SCFAs and also enhance immune system homeostasis in the lungs through supporting the barrier functions of the epithelial cells that are damaged by SEB. We also found that the administration of butyrate (SCFA) alone was able to attenuate ARDS. 

ARDS is a hypoxemic respiratory failure characterized by severe lung inflammatory responses, leading to high mortality that may reach up to 60% [7]. ARDS is triggered by a variety of insults, such as pneumonia, sepsis, trauma, and certain viral infections. SARS-CoV-2 infection, which causes COVID-19, has also been shown to trigger viral pneumonia, leading to ARDS in patients with the severe form of the disease [46]. ARDS can also cause alterations in the microbiota in the lungs, leading to secondary infections. The prevalence of coinfection varies among COVID-19 patients, as indicated in different studies, but it can account for up to 50% among patients who die from COVID-19 [8]. The co-pathogens include *Staphylococcus aureus* [8], which produces SEB and which can also promote ARDS and cytokine storm. Thus, characterization of the lung microbiome is an important aspect of understanding the mechanisms involved in ARDS. SEB can also alter the gut microbiota and thereby alter the lung–gut axis that drives immune homeostasis. It is for these reasons that the current study focused on the role of SEB in altering the microbiota in the lungs and the gut and further addressed the question of whether AEA-mediated attenuation of ARDS is associated with microbiome profile. It should be noted that the effect of AEA on microbiota in the gut and lungs has not been investigated before. 

The results obtained from the current study based on clinical lung parameters were consistent with our previous studies [9] demonstrating that AEA treatment significantly improved all the parameters of ARDS in the lung, including peak inspiratory flow, peak expiratory flow, and tidal volume. In the current study, the expression of IL-6, one of the proinflammatory cytokines, was significantly increased in mice with ARDS, and AEA treatment significantly decreased IL-6 expression in both serum and the bronchoalveolar lavage fluid of mice. IL-6 has been shown to be highly expressed in respiratory infections, including SARS-CoV-2/COVID-19 patients and other respiratory infections [47,48]. 

AMPs are produced primarily by the epithelial cells which play an important role in host defense against infections [49]. AMPs have been applied as a therapeutic agent against lung infections [50]. In the current study, using scRNA-seq, we found that AEA increased the expression of several AMPs by the lung epithelial cells. TAP1 and TAP2 are among the AMPs produced by lung epithelial cells exclusively in the respiratory system that are specifically protective against pathogens, such as Pseudomonas spp. [51,52]. SLPI has also been shown previously to play an important role against pathogenic bacteria, including Pseudomonas, during respiratory infections [53,54]. Interestingly, in the current study we noted that SEB + VEH group had increased the abundance of Pseudomonas spp. while they were absent in the naïve and SEB + AEA groups. These data suggested that SEB may promote the growth of Pseudomonas and that AEA may suppress its growth by inducing AMPS. Additionally, mBD2, which is an antimicrobial peptide, not only inhibits the growth of pathogenic bacteria, such as *Pseudomonas aeruginosa*, but also promotes macrophage functions [55]. In the current study, we noted that AEA treatment increased the expression of mBD2, which was associated with decreased Pseudomonas in the SEB + AEA group when compared to SEB + VEH group. Similarly, LYZ2, another AMP which was induced by AEA has been shown to play a key role in killing bacteria in the lungs and preventing the spread of the infection [56,57]. Additionally, S24_7 (Muribaculaceae) which is a beneficial bacteria [58,59], was seen in the SEB + AEA group in both the gut and the lungs, suggesting that there may be crosstalk between gut and lung microbiomes. Our LEfSe analysis indicated that Enterobacteriaceae and Pseudomonas were both present in the lungs of the SEB + VEH group, but not in the naïve or SEB + AEA groups. It is interesting to note that both of these bacteria are considered to be the underlying cause of pneumonia and lung diseases [60,61]. Additionally, Tenericutes, which is considered to be pathogenic [62], was indicated in the SEB + VEH group only but not in the Naïve or SEB + AEA groups in the gut. Lachnospiraceae and Clostridia, which is one of the main producers of butyrate [63], were indicated after AEA treatment, but not in the SEB + VEH group. Together, these data suggested that SEB may trigger harmful bacteria and that AEA not only prevents these bacteria from spreading but also induces beneficial bacteria in the lungs and the gut. 

Tight junctions play an important role in the intestinal barrier functions which control permeability [64,65]. In the current study, we noted that Claudin and E-cadherin were upregulated by AEA. CLDN1 is one of the main tight junctions that maintains intestinal epithelial homeostasis [66]. A decrease in CLDN1 expression leads to an increase in the leakiness during lung injury [66]. E-cadherin (CDH1) plays a critical role in the epithelial cell barrier functions [67] and loss of E-cadherin function can lead to respiratory disorders [68].

Crosstalk in the gut-lung axis is one of the important pathways for maintaining immune homeostasis [69]. Physiological pathways of the gut-lung axis are thought to be bidirectional and to establish a well-maintained relationship between the gut and the lungs, as well as playing a crucial role in maintaining health and homeostasis [70]. The gut-lung axis has been investigated before. Previous studies from our lab indicated that cannabinoids, specifically THC, were able to modulate the microbiome profile of both the gut and the lungs in a similar way during ARDS [38]. To date, there are no studies showing the immune modulation effect of the endocannabinoid AEA on the microbiome profile of the gut-lung axis. 

In the current study, we noted that some bacteria were common in both the lungs and the guts of naïve mice, such as Betaproteobacteria. Additionally, S24_7 (Muribaculaceae) beneficial bacteria, were seen both in the lungs and the gut of the SEB + AEA group. These findings supported the concept of crosstalk between gut and lung microbiomes [71,72,73]. Our study also indicated that AEA treatment led to a decrease in the abundance of the order Pseudomonas and Caulobacterlaes, which are both believed to cause human respiratory tract infections [38,74]. In the current study, we also noted that there was increased abundance in the gut of the family Anaplasmataceae, whose members are α-proteobacteria, classified in the order Rickettsiales, and which currently contains five genera, some of which contain members that are known to infect humans [75] Anaplasmataceae members are well recognized for causing tick-borne life-threatening zoonotic diseases in the United States [75]. Thus, it was interesting to note that AEA prevented the increase in Anaplasmataceae induced by SEB. We also noted that SEB caused an increase in the phylum Tenericutes in the gut, which consists of wall-less bacteria. Their members establish commensal or highly virulent relationships in animals and humans [76]. In addition, in the lungs, SEB caused an increase in Caulobacteraceae, which are gram-negative proteobacteria that can include a variety of pathogenic bacteria [77], as well as an increase in Pseudomonadales, some members of which are pathogens [78]. Interestingly, AEA reversed the effect of SEB on these bacteria. In the current study, we also noted that AEA treatment increased the abundance of S24_7, currently called Muribaculaceae, a family of bacteria within the order Bacteroidales [58]. These bacteria are normally found in the gut [58], but, interestingly, we also found these in the lungs of AEA-treated mice, which also suggested gut-lung axis and crosstalk. Furthermore, Muribaculaceae are known to promote fermentation pathways to produce SCFAs [79], which were also found to be increased following AEA treatment. 

In this study, analysis of gut microbiomes, as well as LEfSe, demonstrated that AEA treatment increased numbers of Bacteroidetes and Clostridia, which are the main bacteria that produce butyrate [80]. Additionally, we found that AEA induced the abundance of Lachnospiraceae, a family of anaerobic bacteria in the order Clostridiales that ferments diverse plant polysaccharides to SCFAs [81], which were found to be increased in mice treated with AEA. Thus, the gut-lung axis may allow SCFAs to get into the blood stream and into the lungs and to suppress inflammation in the lungs. To that end, our studies also demonstrated that direct administration of butyrate in SEB-injected mice suppressed ARDS. Sodium butyrate is very well known for its anti-inflammatory and immune modulatory properties. 

Lymph nodes are considered the site of immune response initiation against pathogens [82]. Previous studies from our lab have demonstrated the role of MLNs in the modulation of microbiomes [35]. Another important study has shown that mesenteric lymph nodes significantly attenuated inflammatory biomarkers of the lung to preserve barriers in a septic model of rats [83]. Additionally, another study demonstrated that gut-associated lymph nodes serve as a main source for the initiation of systemic inflammatory responses, specifically against acute lung injury [84]. Thus, in the current study, we investigated the immune responses in MLNs, specifically the response of T cells after AEA treatment. AEA significantly decreased the T cell subsets, including CD4 + T cells, CD8 + T cells, Vβ8 + T cells, and NK + T cells in the MLNs compared to the SEB + VEH group. 

It should be noted that there are some limitations in our study. Since we tested ARDS in only female mice, it also needs to be tested whether AEA is effective in male mice. Furthermore, it is important to determine how late AEA can be administered after SEB exposure and to gauge its effectiveness during various stages of ARDS. Additionally, in the current study, while we tested the effect of AEA in ARDS mice, whether it alters the microbiota and SCFA production in normal naïve mice was not studied. This is because our goal was to see if AEA had an effect on microbiomes during ARDS rather than in naïve mice. However, previous studies have shown that naïve C57BL/6 mice injected with AEA congener intraperitoneally had the same microbiome profiles as naïve control C57BL/6 mice, as both groups clustered together in PCOA plots, suggesting that their microbial compositions were similar [85]. Additionally, our lab reported previously that naïve C57BL/6 mice injected with THC alone, which is in the same class of cannabinoids and considered a mimic to AEA and a CB1 agonist, did not exhibit significant alterations in the microbiota [86]. In the current study, we did not investigate the effect of butyrate in naïve mice. Our laboratory has reported previously that naïve C57BL/6 mice treated with butyrate alone did not exhibit significant differences in gut microbial composition [87]. 

In summary, the current study demonstrates that SEB, a bacterial superantigen, induces ARDS, triggering inflammation not only in the lungs but also in the gut. Interestingly, SEB also caused alterations in the lung and gut microbiota, specifically causing an abundance of pathogenic bacteria, such as Pseudomonas, while AEA reversed this effect. AEA induced several antimicrobial peptides and it is likely that this effect may have prevented the emergence of pathogenic bacteria following SEB treatment. AEA also promoted the abundance of beneficial bacteria that produce SCFAs, such as butyrate. Thus, AEA may attenuate SEB-mediated ARDS not only through direct suppression of inflammation in the lungs but also by altering the microbiota in the lungs and the gut. Our studies also suggest that increasing endogenous AEA through the use of inhibitors, such as fatty acid amide hydrolase (FAAH), may serve as an effective treatment modality against ARDS. 

## Figures and Tables

**Figure 1 cells-10-03305-f001:**
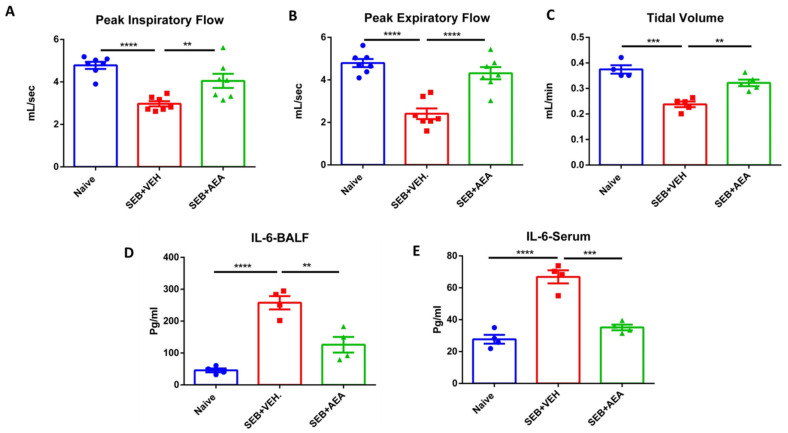
AEA improves the clinical symptoms of ARDS in the lungs induced by SEB. On day −1, 0, and 1, mice received 40 mg/kg of AEA or VEH I.P. and SEB at a dose of 50 μg/mouse intranasally on day 0. Mice were euthanized 48 h after SEB exposure. The lung functions were assessed using plethysmography. The data shown include clinical functions of the lung, including (**A**) peak inspiratory flow, (**B**) peak expiratory flow, and (**C**) tidal volume (TV). (**D**) Measurement of cytokines IL6 in the sera and (**E**) bronchoalveolar lavage fluid (BALF). In panels (**A**–**C**), seven mice were used in each group, and in panels (**D**,**E**) four mice were used in each group. One-way ANOVA with post hoc Tukey’s test was used to compare the three groups. The data were confirmed in three independent experiments. The vertical bars represent Mean ± SEM and statistical significance was indicated as follows: * *p* ≤ 0.05, *p* ** ≤ 0.01, *** *p* ≤ 0.001, **** *p* ≤ 0.0001.

**Figure 2 cells-10-03305-f002:**
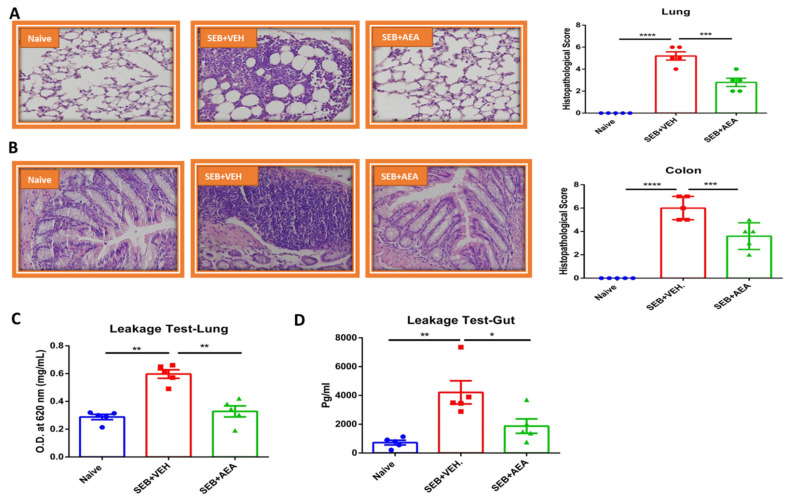
AEA decreases inflammatory parameters of the gut and lungs. Mice were treated with SEB and AEA as described in the legend for Figure 1. (**A**) Representative histopathological images and scores. Images were generated by using Keyence microscope with 40x magnification planAPO NA0.95 and H&E staining of excised lung tissue from Naïve, SEB + VEH and SEB + AEA groups of mice. (**B)** Representative histopathological images and scores.Images were generated by using Keyence microscope with 40x magnification planAPO NA0.95 with H&E staining of colon tissue from Naïve, SEB + VEH and SEB + AEA groups of mice. (**C**) Gut leakage in the gut. (**D**) Capillary leakage in the lung of Naïve, SEB + VEH, and SEB + AEA mice. Five mice were used in each group and the data were confirmed in three independent experiments. One-way ANOVA with post hoc Tukey’s test was used to compare the three groups. The vertical bars represent mean ± SEM and statistical significance was indicated as follows: * *p* ≤ 0.05, ** *p* ≤ 0.01, *** *p* ≤ 0.001, **** *p* ≤ 0.0001.

**Figure 3 cells-10-03305-f003:**
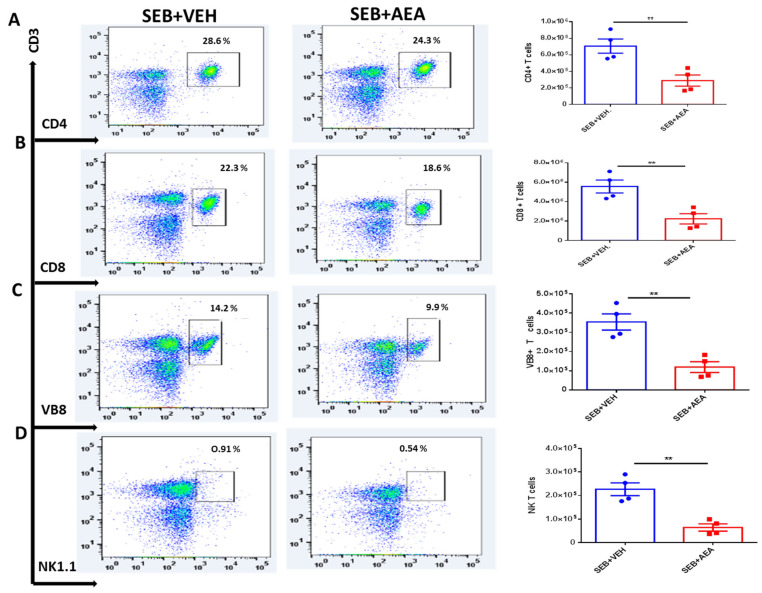
AEA decreases T cell subpopulations in the MLN. Mice were treated with SEB and AEA as described in the Figure 1 legend. Each panel shows a representative experiment depicting MLN analyzed for percentages of various T cell markers. The data on total cellularity from four mice per group is presented in the form of vertical bars with mean +/− SEM. (**A**) CD3 + CD4 + T cells. (**B**) CD3 + CD8 + T cells. (**C**) CD3 + Vβ8 + T cells. (**D**) CD3 + NK1.1 + cells. Data presented are representative of three independent experiments. A Student’s *t*-test was used to compare the two groups and the statistical significance was indicated as follows: ** *p* ≤ 0.01.

**Figure 4 cells-10-03305-f004:**
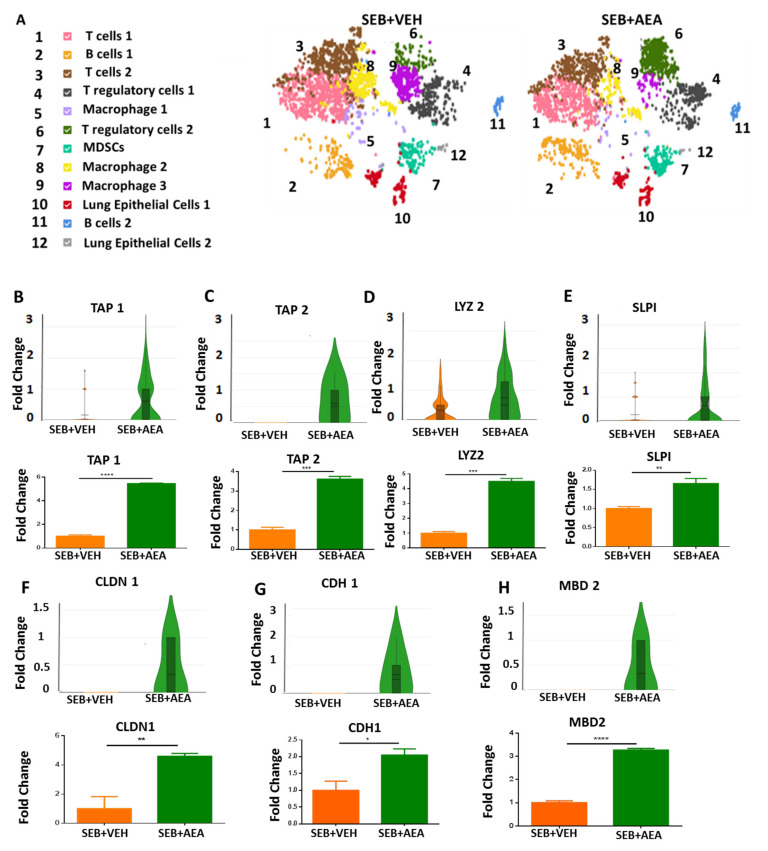
AEA-mediated induction of AMPs and tight junction proteins during ARDS, analyzed using scRNA-seq of the lungs and RT-qPCR. Mice were treated with SEB and AEA, as described in the Figure 1 legend. (**A**) Different types of cell clusters in the lungs of SEB + VEH and SEB + AEA mice. The cells were classified based on markers as follows: Panel (**A**) (1) T cells 1 (CD3+, CD8+, CD44+, Trbc1), (2) B cells 1 (CD19+, CD38, PAX5), (3) T cells 2 (CD3+, Trbc2, CD8+, IFNG, Trbc1), (4) T regulatory cells 1 (CD4+, SATB1+, FOXP3+, MAF+, SELL+, CLTA4+, STAT5B, IKZf2, Lag3, Nt5e), (5) Macrophage 1 (F4/80, Fth, Ftl, CD68), (6) T regulatory cells 2 (CD4+, SATB1, MAF+, SELL+, CLTA4+, STAT5B, IKZf2, CCr4, Gata3), (7) Myeloid Derived Suppressor Cells (MDSCs) (CD11B+, GR1+, ARG1+, LY6C+), (8) Macrophage 2 (F4/80, Fth, Ftl, Siglec F), (9) Macrophage 3 (F4/80), (10) Lung Epithelial cells 1 (TAP1+, TAP2+, SLPI+, Lyz2+, MBD2, LTF, CDH1, CLDN1), (11) B cells 2 (CD19+, PAX5), (12) Lung Epithelial cells 2 (TAP2+, SLPI+, Lyz2+, MBD2, EPCAM+, CDH1). Panels (**B**–**H**) represent various molecules studied on epithelial cells from the lungs using scRNA-seq (upper panels) and validated using RT-qPCR (lower panels). Single-cell data were obtained by pooling cells from five mice per group and running 3000 total cells per group. To validate these results, RT-qPCR data were obtained from five mice per group. The vertical bars represent mean ± SEM. The statistical significance was analyzed using a Student’s *t*-test and indicated as follows: * *p* ≤ 0.05, ** *p* ≤ 0.01, *** *p* ≤ 0.001, **** *p* ≤ 0.001.

**Figure 5 cells-10-03305-f005:**
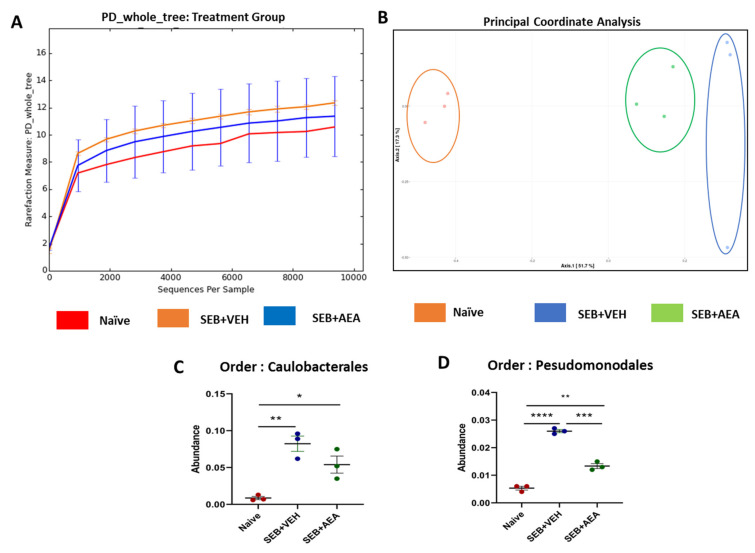
The role of AEA on the abundance of microbiota in the lungs of ARDS induced by SEB. Mice were treated with SEB and AEA as described in the Figure 1 legend. The lungs were collected and 16S rRNA sequencing and the Nephele platform were used to analyze and generate: (**A**) Rarefaction curves depicting alpha diversity within groups (Chao1 index); (**B**) Principal coordinate analysis which reveals the clustering of bacteria in lungs based on their 16S rRNA content similarity. GraphPad Prism was used to analyze the abundancy of microbiome in panels (**C**,**D**). The data represent three mice per group. One-way ANOVA with post hoc Tukey’s test was used to compare the three groups. Statistical significance was indicated as follows: * *p* ≤ 0.05, ** *p* ≤ 0.01, *** *p* ≤ 0.001, **** *p* ≤ 0.0001.

**Figure 6 cells-10-03305-f006:**
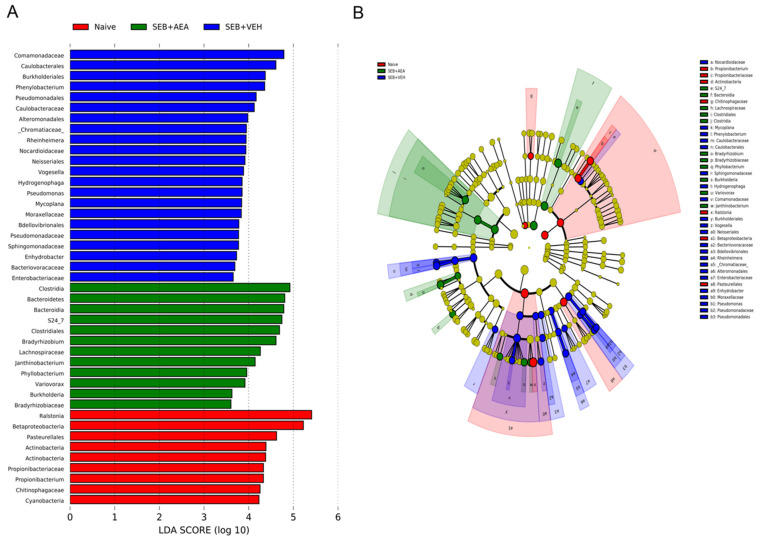
Linear discriminant analysis of effect size (LEfSe) in the lungs. (**A**) LEfSe-generated linear discrimination analysis (LDA) scores for differentially expressed taxa. (**B**) LEfSe-generated cladogram for operational taxonomic units (OTUs) showing phylum, class, order, family, genus, and species from outer to inner swirl. For LEfSe data, the alpha factorial Kruskal–Wallis test among classes was set to 0.05, and the threshold on the logarithmic LDA score for discriminative features was set at 3.

**Figure 7 cells-10-03305-f007:**
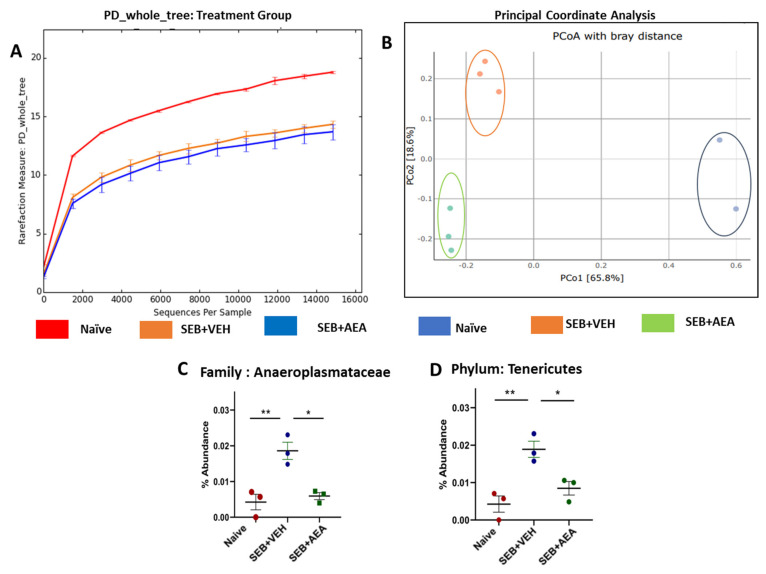
Effect of AEA on the abundance of microbiota in colon/cecal flush of mice with SEB-mediated ARDS. Mice were treated with SEB and AEA, as described in the Figure 1 legend. The samples of cecal flush were collected and, using 16S rRNA sequencing and the Nephele platform, the data were analyzed. (**A**) Rarefaction curves depicting alpha diversity within groups (Chao1 index). (**B**) Principal coordinate analysis which reveals the clustering of bacteria in lungs based on their 16S rRNA content similarity. GraphPad Prism was used to analyze the abundancy of microbiome in panels (**C**,**D**). Data in panels (**C**,**D**) represent mean ± SEM from three mice/group. While three mice were also used in panel B, the sequence reads for one of the samples in the naïve group did not meet the threshold for PCOA, thereby showing two samples, while this sample qualified enough for the Chao index, abundancy, and LEfSe analysis. Significant differences between groups were calculated using one-way ANOVA with post hoc Tukey’s test. Significance is indicated as follows: * *p* < 0.05, ** *p* < 0.01.

**Figure 8 cells-10-03305-f008:**
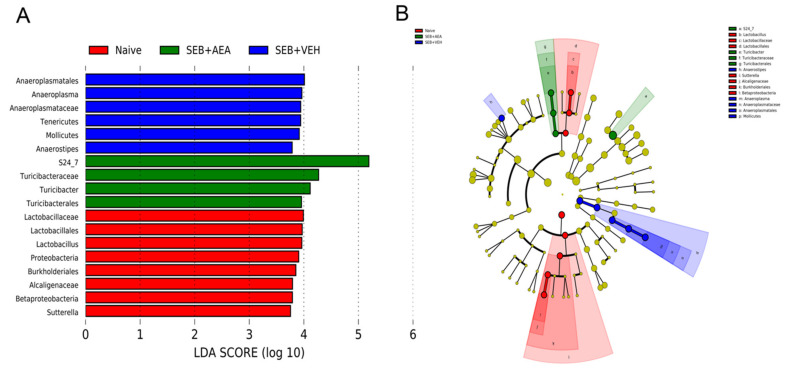
Linear discriminant analysis of effect size (LEfSe) in the cecal flush of mice with ARDS. Mice were treated with SEB and AEA, as described in the Figure 1 legend. (**A**) LEfSe-generated linear discrimination analysis (LDA) scores for differentially expressed taxa. (**B**) LEfSe-generated cladogram for operational taxonomic units (OTUs) showing phylum, class, order, family, genus, and species from outer to inner swirl. For LEfSe data, the alpha factorial Kruskal–Wallis test among classes was set to 0.05, and the threshold on the logarithmic LDA score for discriminative features was set at 3.

**Figure 9 cells-10-03305-f009:**
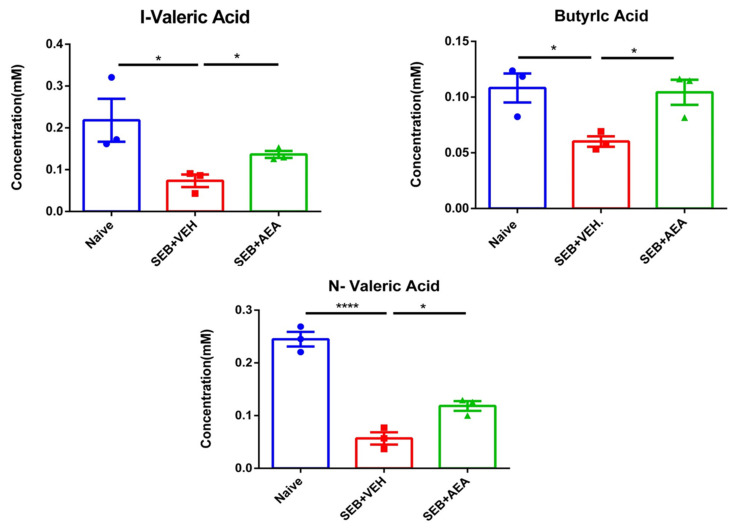
Analysis of SCFAs from cecal flushes of mice with ARDS. Mice were treated with SEB and AEA, as described in the Figure 1 legend. Concentrations of SCFAs from the cecal flushes were measured. Vertical bars represent mean ± SEM from three mice/group and significant differences between groups were analyzed using one-way ANOVA with post hoc Tukey’s test. The significant differences are shown as follows: * *p* < 0.05, **** *p* < 0.0001.

**Figure 10 cells-10-03305-f010:**
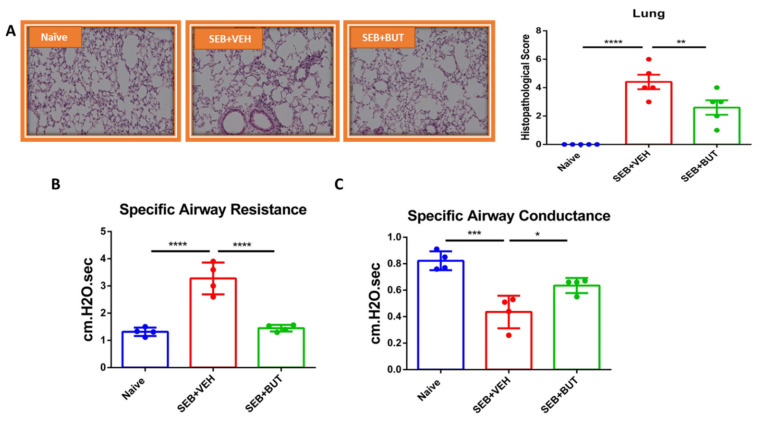
Effect of butyrate on ARDS induced by SEB. On day −1, 0, and 1, mice received 200 mg/kg of butyrate (BUT) or VEH through oral gavage, while SEB at a dose of 50 μg/mouse was given intranasally on day 0. Mice were euthanized on day 2 for various studies (48 h after SEB exposure). (**A**) Representative histopathological images and scores.Images were generated by using Keyence microscope with 40x magnification planAPO NA0.95 and H&E staining of excised lung tissue from Naïve, SEB + VEH and SEB + BUT groups of mice. (**B**,**C**) Use of plethysmography to measure the clinical function of the lung, including (**B**) specific airway resistance and (**C**) specific airway conductance. Five mice in each group for histopathology and four mice per group were used for the experiment on plethysmography, and the results were confirmed in three independent experiments. One-way ANOVA with post hoc Tukey’s test was used to analyze for significance and was indicated as follows: * *p* ≤ 0.05, ** *p* ≤ 0.01, *** *p* ≤ 0.001, **** *p* < 0.0001.

**Table 1 cells-10-03305-t001:** The details of Primer sequences used for RT-qPCR.

Name of the Gene	Primer	Sequence
Tracheal Antimicrobial Peptide 1(TAP1)	Forward Reverse	5-GGA CTT GCC TTG TTC CGA GAG-3 5-GCT GCC ACA TAA CTG ATA GCG-3
Tracheal Antimicrobial Peptide2 (TAP2)	Forward Reverse	5-CTC CCA CTT TTA GCA GTC CCC-3 5-CTG GCG ATG GCT TTA CTT-3
Secretory Leukocyte Peptidase Inhibitor (SLPI)	Forward Reverse	5-TAC GGC ATT GTG GCT TCT CAA-3 5-TAC GGC ATT GTG GCT TCT CAA-3
Lysozyme 2	Forward Reverse	5-ATG GAA TGG CTG GCT ACT ATG-G 3 5-ACC AGT ATC GGC TAT TGA TCT GA-3
Claudin 1 (CLDN1)	Forward Reverse	5-GGG GAC AAC ATC GTG ACC G-3 5-AGG AGT CGA AGA CTT TGC ACT-3
Epithelial Cadherin (ECDH)	Forward Reverse	5-CAG GTC TCC TCA TGG CTT TGC -3 5-CTT CCG AAA AGA AGG CTG TCC-3
Murine Beta Defensine 2 (MBD2)	Forward Reverse	5-AGA ACA AGG GTA AAC CAG ACC T-3 5-ACT TCA CCT TAT TGC TCG GGT-3

## Data Availability

The datasets presented in this study can be found in online repositories: https://www.ncbi.nlm.nih.gov/sra. Accessed on 22 November 2021 and 24 November 2021 with submission IDs SUB10704732, SUB10704983 and BioProject IDs PRJNA782645, PRJNA783036 respectively.

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
