# Peer review of "Endocannabinoid Anandamide Attenuates Acute Respiratory Distress Syndrome through Modulation of Microbiome in the Gut-Lung Axis"

_cells, 2021, doi:10.3390/cells10123305_

Round 1
Reviewer 1 Report
A manuscript „Endocannabinoid Anandamide attenuates Acute Respiratory 2 Distress Syndrome Through Induction of Antimicrobial Pep- 3 tides Leading to Modulation of Microbiome Profile in Gut– 4 Lung Axis and Driving the Immune Homeostasis“ is hard to follow.
Correct formal mistakes lines 33-34, or double/non-spaces for example (Nagarkatti et al., 2009;Cristino et al., 2020).
Line 100, reference sultan et al., 2021 is inappropriate because in this article there is no description of ARDS induction. Add better references.
The methods should be more detailed, it is quite confusing.
Correct BALF = Bronchoalveolar lavage fluid
How many animals were used in each group? 3,5 or 7?
Data should be presented as individual data dots.
Improve the quality of pictures.
Add limitation of the study.
Consider the analysis of oxidative stress parameters and the effect of anandamide.
Reviewer 2 Report
In a previous article, the authors demonstrate that AEA can effectively and partially suppress the damage caused by SEB-induced ARDS in mice.
In the present study, the authors attempt to demonstrate that this effect could be related to the increase in SCFA produced by beneficial bacteria.
Beneficial bacteria would increase because of AEA.
The authors demonstrate the reciprocal relationship of the intestinal and pulmonary microbiota during infection.
Also, that the effect of AEA in case of infection follows this same pattern. Specifically, Figures 6 and 8 are very illustrative.
Despite the quality and interest of the study, I believe that before publication it is necessary to have a new control to see how the microbiota changes (or not) because of AEA in the absence of infection.
Likewise, how the profile of the SCFAs is modified if it does. Therefore, it should be necessary to include a control of mice that only receive AEA.
Reviewer 3 Report
In this study authors examined the effect of anandamide on Staphylococcal enterotoxin B-mediated ARDS using mouse animal model. Extensive and detailed research has been performed, included assessment of * the level of Anti-Microbal Peptides in lung epithelial cells, * advancement of inflammatory process in lungs and gut-associated mesenteric lymph nodes as well as * alterations of gut and lungs microbiota.
The strength of the paper is the multidirectional assessment of the effect of anandamide in ARDS on the microbiome and the lung-gut axis. These are issues of interest to both scientists and clinicians.
The weakness of the manuscript is that it presents the results only for female mice and in a very limited number of animals. Mortality from ARDS, especially from COVID-19 infection, is more than 2 times higher in men. There are sex-specific differences in the gut microbiome and the gut-lung axis. In addition, the microbiome is closely related to the immune system, which changes in females during the cycle.
There are several issues that, in my opinion, must be addressed in order for this manuscript to be suitable for publication:
- How was the group size calculated?
- There is no description of the size of the groups in the Method section.
- Why was the experiment performed only on females?
- In which phase of cycle were the females during the experiment? Such information should be included in the methodology. It should also be noted in the Abstract and Discussion section that the experiment was limited to studies on females only.
- How was the anandamide dose selected? Please enter the name of the manufacturer.
- Enter the values for the F and t statistics with df in the Results section.
- Fig.1- panels D and E shows the results for 4 animals, as described there should be 7 mice / group.
Fig.2-panel C – no unit on the Y axis
Fig.3- The figure shows the results for 4 animals, as described there should be 5 animals. Have the missing results been removed?
Fig.4- I propose to split figure 4 into two separate ones. Now it does not fit on the page. The font on Figure 4a is illegible.
Fig.6 and 8- The font is completely illegible.
Fig.7B - There were only 2 animals in the naive group?
Fig.9- correct units of concentrations
Fig.10 B and C-the figures show the results for 4 animals, while according to the description it should be 5/group
- 457-459: “SLPI which has also been shown….” the sentence seems to be unfinished. Correct it.
- 473-476: “Our Lefse analysis…” paraphrase this sentence, now it’s unclear.
- 502-503: “These findings supported …” – I suggest you add the appropriate references, eg. doi: 10.3389/fmicb.2020.00301. eCollection 2020.
doi: 10.1513/AnnalsATS.201503-133AW.
doi: 10.1128/JB.00454-20. Print 2021 Jan 25.
- At the beginning of the discussion, I suggest to emphasize the innovative nature of the research, the lack of results available in the literature and the most important discoveries of this project.
Reviewer 4 Report
Dear Authors, I found your manuscript interesting and pleasant to read. Your experiment and your research are very detailed and well conducted.
In following items I summarize my suggestions and requests before editing:
- I think that could be better to insert in “Material and Methods” a report about chemical and reagent purchase. If it was equal to that published in your recent paper (Sultan M et al 2021), it would be sufficient a reference citation.
- I think that the dosing’s time of AEA is a “Key point” in your experiment. In effect such an early administering (on day 1 after SEB exposure) could be very unusual in clinical practice. In my opinion future studies should address the timing AEA dosing to see how long after SEB exposure, its delivery is still efficacious and useful.
- Line 252 you should insert “(E)” in your explanation of fig. 1 when you relate about IL-6-BALF measure.
- Line 263: you could insert a short explanation about the Histopathological score, or some references for a better understanding.
- Line 354: I have following little request for Authors and Editorial Office: enlarge Figure 6A and 6B, and Figure 8B to improve understanding.
- Line 415 Fig 10A, better correct ml/min as airway resistances’ unit of measure. In effect ml/min is "minute ventilation" unit of measure
Best regards
Round 2
Reviewer 2 Report
After analysing the changes in the manuscript and the clarifications made by the authors, I don't see that the problems I had raised have been resolved.
With respect to point 1, I do not see that the results of the article of Guida et al. (Altered gut microbiota and endocannabinoid system tone in vitamin D deficiency-mediated chronic pain) justifies that the microbiome is not altered by injection of cannabinoid receptors agonists.
On the other hand, this paper is not even cited in the manuscript of Guida et al. to justify the lack of controls.
Likewise, the authors indicate that they are working on another independent project where we study the microbiome profile in FAAH KO mice (Fatty Acid Amide Hydrolyse) and indicate that these mice have high levels of AEA and want to know if they have the altered microbiota. This is very interesting and complementary, but it is not part of the manuscript and the reader will ignore the results of this parallel research.
With respect to point 2. The authors state in the abstract: "AEA triggered several bacterial species that produced increased levels of Short Chain Fatty Acids (SCFAs), including butyrate." They also indicate that SCFAs stabilize the gut-lung microbial axis driving the immune homeostasis. They also state that “that administration of butyrate (SCFA) alone was able to attenuate ARDS”. Finally, they indicate: “Also, in the current study, while we tested the effect of AEA in ARDS mice, whether it alters the microbiota and SCFA production in normal naïve mice was not studied.”
Although the authors’ response indicates that they know more about these aspects and that some are working on it, all this is not reflected in the manuscript. Therefore, I believe that before publication it is necessary to have a new control to see how the microbiota changes (or not) because of AEA in the absence of infection. Also, a new control relative to SCFAs changes because of AEA in the absence of infection.
Author Response
Dear Reviewer,
Please see the response in the attached file.

Reviewer 3 Report
I consider that the revised version of the present manuscript improved significantly regarding the previous version.
The authors answered all the questions raised and made the appropriate changes in the manuscript.
I consider the manuscript is now suitable for publication.
Author Response
Dear Reviewer,
We would like to thank you for your valuable suggestions, comments, and your time.
Very Sincerely,
Muthanna Sultan